# Virulence Factors in Colorectal Cancer Metagenomes and Association of Microbial Siderophores with Advanced Stages

**DOI:** 10.3390/microorganisms10122365

**Published:** 2022-11-30

**Authors:** Nour El Houda Mathlouthi, Aicha Kriaa, Leila Ammar Keskes, Moez Rhimi, Radhouane Gdoura

**Affiliations:** 1Laboratoire de Recherche Toxicologie Microbiologie Environnementale et Santé (LR17ES06), Faculté des Sciences de Sfax, Université de Sfax, Sfax 3000, Tunisia; 2Microbiota Interaction with Human and Animal Team (MIHA), Micalis Institute-UMR1319, Ag-roParisTech, Université Paris-Saclay, INRAE, F-78350 Jouy-en-Josas, France; 3Laboratoire de Recherche de Génétique Moléculaire Humaine, Faculté de Médecine de Sfax, Avenue Majida BOULILA, Université de Sfax, Sfax 3029, Tunisia

**Keywords:** colorectal cancer, gut microbiota, pathogens, virulence factors, siderophores, prognosis

## Abstract

Colorectal cancer (CRC) is a growing public health challenge, featuring a multifactorial etiology and complex host–environment interactions. Recently, increasing evidence has pointed to the role of the gut microbiota in CRC development and progression. To explore the role of gut microbes in CRC, we retrieved metagenomic data from 156 stools from the European Nucleotide Archive database and mapped them against the VFDB database for virulence factors (VFs). GO annotations of VFs and KEGG pathways were then performed to predict the microbial functions and define functional pathways enriched in the tumor-associated microbiota. Interestingly, 306 VFs were detected in the metagenomic data. We revealed the enrichment of adenomas with VFs involved in cell adhesion, whereas in the early stages of CRC they were enriched in both adhesins and isochorismatase. Advanced stages of CRC were enriched with microbial siderophores, especially enterobactin, which was significantly associated with isochorismate synthase. We highlighted higher abundances of porins and transporters involved in antibiotic resistance and the development of biofilm in advanced stages of CRC. Most VFs detected in CRC, particularly in advanced stages, were shown to be included in siderophore biosynthesis pathways. This enrichment of predicted VFs supports the key role of the gut microbiota in the disease.

## 1. Introduction

In 2020, the International Agency for Research on Cancer recorded 935,173 deaths caused by colorectal cancer (CRC), making it the third most common cancer [1]. The etiology of CRC involves both genetic and environmental factors [2]. Inherited cancers account for only 5% of all CRC cases and could be classified into polyposis (familial adenomatous polyposis (FAP)) and hereditary nonpolyposis colorectal cancer (HNPCC) [3]. Unlike these cancers which are driven by inherited mutations [4], sporadic cancers stemming from point mutations in the lifetime, target several loci linked to the initiation of the adenoma formation and the development and progression of carcinoma [5]. In addition, personal medical history of inflammatory bowel disease (IBD) could promote carcinogenesis [6]. Another important risk factor was strongly associated with CRC is lifestyle, including dietary patterns (a diet high in meat and low in fiber [7]), obesity [8], physical inactivity [9], and cigarette smoking [10].

The role of the gut microbiota in CRC is increasingly recognized [11]. Accordingly, we have recently witnessed the emergence of the holobiont concept to understand the host-microbiota interplay and its relevance in the disease [12]. Next-generation sequencing technologies have advanced our knowledge of the human microbiota as they have enabled the characterization of noncultivable microbes, among others, in health and disease and the prediction of their function [13]. Bioinformatic analysis presents one of the most important steps in metagenomic research; it translates a large amount of biological data into interpretable results. One of the many crucial analyses that we can cite is profiling the taxonomic composition of microbial communities and mapping it against a database such as the Virulence Factor Database (VFDB), which is a reference database for virulence factors produced by bacterial pathogens [14].

Indeed, pathogenic bacteria can produce virulence factors (VFs) that could affect different signaling pathways and induce colorectal carcinogenesis [15]. VFs could be categorized into several classes based on their functions: adherence and colonization factors, invasion factors, capsules and other surface components, and endotoxins [16]. It has been previously shown that the virulence factor of enterotoxigenic *Bacteroides fragilis* (ETBF) and *Fusobacterium adhesin* A (*Fad*A) can bind to intestinal epithelial cell receptors and cleave the extracellular domain of E-cadherin (anchoring junction protein), causing a loss of cell–cell contact and leading to the activation of the NF-kB pathway [17,18]. Other virulence factors such as colibactin and cytolethal distending toxins can trigger host cell double-strand DNA breaks. These factors could alter the immune response of the host and induce the production of pro-inflammatory cytokines [19]. The cytotoxic necrotizing factor 1 (CNF1) produced by *Escherichia coli* leads to cellular motility [20] and enhances proliferation due to the perturbation of the host cellular cycle [21].

In the present study, we aim to uncover the association of virulence factors produced by microbial pathogens in CRC patients at different stages in comparison to healthy subjects by analyzing published metagenomic data generated by whole-genome sequencing from stools of healthy subjects and CRC patients. Next, we will perform a functional profiling of the virulence factors of interest as well as pathway analysis. The overall objective is not only to understand the role of pathogens in colorectal carcinogenesis but also to reveal the importance of the virulence factors in CRC.

## 2. Materials and Methods

### 2.1. Published Fecal Metagenomes

We retrieved gut metagenomic data of the project PRJEB6070 from the European nucleotide archive. This project involves 156 fecal samples from the French population, including 53 from CRC patients (15 from stage I, 7 from stage II, 10 from stage III, and 21 from stage IV), 42 samples from adenoma patients (15 from large adenoma and 27 from small adenoma), 61 samples from randomly selected controls, and 43 samples from the German population (38 fecal samples from CRC patients and 5 samples from healthy subjects). These 199 samples were sequenced using two different techniques: whole-genome shotgun and 16S rRNA gene sequencing. We only analyzed the whole metagenomic data from 156 French samples [22]. (Appendix A). Datasets of published fecal metagenomes are available from the European Nucleotide Archive (ENA) database (https://www.ebi.ac.uk/ena/browser/home, accessed on 25 October 2022), accession number: ERP005534.

### 2.2. Detection of Virulence Factors (VFs)

In order to detect the virulence factors of bacterial pathogens, we mapped the reads against the virulence factor database (VFDB) [14]. The mapping was realized using tools from PATRIC website (https://patricbrc.org/app/MetagenomicReadMapping) accessed from 9 December 2021 to 6 January 2022 [23,24].

### 2.3. Statistical Analysis

The Chi-square test and Spearman correlation coefficient were calculated to estimate the correlations between cancer progression and the detected VFs.

Multivariate analysis was performed to investigate the linear relationship between Isochorismate synthase as a dependent variable and type 1 Fimbriae regulatory protein (FimB), type 1 Fimbriae regulatory protein FimE, type 1 Fimbrial protein A chain precursor, iron-enterobactin ABC transporter permease, phosphopantetheinyl transferase component of enterobactin synthase multienzyme complex, FimG protein precursor, ferrienterobactin ABC transporter periplasmic binding protein, *E. coli* common pilus usher EcpC, enterobactin/ferric enterobactin, esterase, enterobactin exporter (iron-regulated), general secretion pathway protein D, general secretion pathway protein K, and aerobactin synthesis protein IucB as covariates.

Statistical analyses were performed using statistical software package SPSS 20.0. (SPSS 20.0 for Windows; SPSS Inc., Chicago, IL, USA). A two-tailed *p* value below 0.05 was considered statistically significant.

### 2.4. Functional Profiling of VFs, Pathway Analysis, and Association of VF’s Functions with Cancer Progression

Correlated VFs with cancer progression were mapped against the Uniprot–Gene Ontology (GO) database (https://www.uniprot.org/help/gene_ontology) accessed from 9 December 2021 to 6 January 2022, and the functional annotation pathway mapping of VFs was realized using KEGG pathway (https://www.kegg.jp/kegg/pathway.html) they were accessed from accessed from 20 December 2021 to 10 January 2022. A Venn diagram was used to display the most detected functions of VFs correlated with each CRC stage compared to healthy state (http://bioinformatics.psb.ugent.be/webtools/Venn/) accessed on 25 January 2022.

## 3. Results

### 3.1. Published Fecal Metagenomes

By mapping the reads against the Virulence Factor Database (VFDB), we detected the presence of 306 VFs (Appendix A). These factors are produced by pathogenic microbial species that belong to different phyla: Firmicutes (*Clostridium perfringens*, *Enterococcus faecalis*, *Enterococcus faecium*, *Streptococcus agalactiae* and *Streptococcus pneumoniae*), *Actinobacteria* (*Corynebacterium diphtheriae*), and *Proteobacteria* (*Escherichia coli*, *Haemophilus influenzae*, *Klebsiella pneumoniae*, *Pseudomonas aeruginosa*, *Salmonella enterica*, *Shigella dysenteriae*, *Shigella flexneri* and *Yersinia pestis*).

### 3.2. VFs as Markers of Cancer Progression: Bivariate Analysis

Correlation analysis was performed to explore the virulome of different CRC stages and investigate the link between the VFs from pathogenic features and the disease stage (see Appendix A).

The comparison of different stages of CRC with the normal state revealed an enrichment of *K. pneumoniae* VFs in small adenoma, and *E. coli* VFs and *K. pneumoniae* in large adenoma (*p* < 0.05). *H. influenzae* VFs and *K. pneumoniae* VFs were more abundant in the early stages of CRC. The advanced stages of CRC were found to be more enriched with *E. coli*, *K. pneumoniae*, *S. flexneri* and *Y. pestis* VFs (Appendix A). The comparison of VF levels at different stages did not reveal any significant correlation between the virulomes of small adenoma vs. large adenoma and large adenoma vs. early-stage CRC. On the contrary, a comparison of adenoma (large and small) with early-stage CRC revealed a significant enrichment with *K. pneumoniae* VFs, *E. coli* VFs and *E. faecalis* VFs in early-stage CRC. We also compared early-stage and advanced-stage CRC and we noticed that stages III and IV were more enriched with *E. coli* VFs, *K. pneumoniae* VFs especially colibactin, *Y. pestis* VFs and *E. faecalis* VF (Appendix A).

### 3.3. Multivariate Analysis

Multivariate analysis revealed that isochorismate synthase 1 was positively correlated with iron-enterobactin ABC transporter permease (*p* < 0.001), FimG protein precursor (*p* = 0.030), *E. coli* common pilus usher EcpC (*p* < 0.001), enterobactin exporter (iron-regulated) (*p* < 0.001), and general secretion pathway protein D (*p* = 0.030), and negatively correlated with type-1 Fimbrial protein, A chain precursor (*p* = 0.014), the phosphopantetheinyl transferase component of enterobactin synthase multienzyme complex (*p* = 0.036), and general secretion pathway protein K (*p* = 0.002) (Table 1).

### 3.4. Functional Annotation of VFs

We found that only VFs involved in cell adhesion were enriched in small adenomas. Large adenoma was enriched with VFs involved in DNA integration, DNA recombination, the cellular response to DNA damage stimulus, cell adhesion, regulator activity, biosynthetic processes, and metabolic processes. As for early-stage and advanced-CRC, the analysis revealed that the fecal microbiome contains VFs that seem to be a part of different processes and activities (Table 2).

To identify relevant functions in CRC stages, we used a Venn diagram (Figure 1). VFs involved in cell adhesion were detected in all stages of CRC, and those involved in the biosynthetic and metabolic processes were found in large adenoma and all stages of CRC. Large adenoma and advanced-stage CRC were enriched in VFs involved in DNA integration, DNA recombination, the cellular response to DNA damage stimulus, and regulator activity. All CRC samples exhibited VFs that play a role in the binding, signal transduction, the regulation of transcription, porin activity, and transportation. Pilus assembly was the only function included exclusively in early-stage CRC. Ligase activity, the NAD metabolic process, protein phosphopantetheinylation, biofilm development, host cell recognition, mechanosensory behavior, the homeostasis process, the detection of virus, viral entry into host cell, response to antibiotics, oxidoreductase activity, and signaling receptor activity were mostly associated with VFs found in advanced-stage CRC.

Functions of targeted VFs detected:-Cell adhesion in large adenoma, small adenoma, CRC I -II, and CRC III-IV;-Metabolic process and biosynthetic process in large adenoma, CRC I -II, and CRC III-IV;-DNA recombination, DNA integration, regulator activity, and cellular response to DNA damage stimulus in large adenoma and CRC III-IV;-Binding activity, the regulation of transcription, porin activity, transporters, and signal transduction in CRC I -II and CRC III-IV;-Pilus assembly in CRC I -II;-Response to an antibiotics, biofilm development, NAD metabolic process, viral entry into the host cell, homeostasis process, oxidoreductase activity, detection of virus, signaling receptor activity, mechanosensory behavior, ligase activity, protein phosphopantetheinylation, and host cell recognition only in CRC III-IV.

### 3.5. Pathway annotation of VFs

The pathway analysis was conducted on the VFs considered to be associated with each stage of CRC (Figure 2). The predominantly detected VFs in small adenoma and large adenoma vs. normal (*p* < 0.05) were not included in KEGG pathway analysis. We found that VFs associated with advanced- and early-stages CRC were involved in many pathways (Table 2).

A Venn diagram was used at this level to explore the relevant pathways in CRC stages. For early CRC, streptomycin biosynthesis and a two-component system were mostly revealed. As for advanced CRC, many pathways were identified, including ABC transporters, ubiquinone and terpenoid-quinone biosynthesis, a bacterial secretion system, pentose and glucuronate interconversions, ascorbate and aldarate metabolism, amino sugar and nucleotide sugar metabolism, the biosynthesis of cofactors, galactose metabolism, starch and sucrose metabolism, and microbial metabolism in diverse environments.

-Pathways that include targeted VFs detected in all CRC stages: O-antigen nucleotide sugar biosynthesis, metabolic pathways, biosynthesis of secondary metabolites, biosynthesis of nucleotide sugars, biosynthesis of siderophore group nonribosomal peptides;-Pathways that include targeted VFs detected in CRC I-II: streptomycin biosynthesis and two-component system;-Pathways that include targeted VFs detected in CRC III-IV: ABC transporters, ubiquinone and terpenoid-quinone biosynthesis, a bacterial secretion system, pentose and glucuronate interconversions, ascorbate and aldarate metabolism, amino sugar and nucleotide sugar metabolism, the biosynthesis of cofactors, galactose metabolism, starch and sucrose metabolism, and microbial metabolism in diverse environments.

## 4. Discussion

Several studies have highlighted the role pathogenic bacteria and their virulence factors in colorectal carcinogenesis [11,15,22]. Defining a clear set of virulence factors as potential targets of interest and investigating the involved pathways is a promising approach for CRC therapy. The analysis of this association requires cohort studies and comparisons of the results in a large-scale meta-analysis [22]. In the present study, we analyzed shotgun-metagenomic data available on ENA, generated from the sequencing of 156 samples in order to explore the virulence factors present in the intestinal microbiota of patients with CRC. Only Fimbriae virulence factors produced by K. pneumoniae mrkD and mrkF were found to be associated with small adenoma. Previous studies revealed their involvement in the surface stability of Fimbriae and facilitation of binding to collagen, respectively [25]. Similarly, large adenoma seemed to be significantly associated with Fimbriae virulence factors produced not only by K. pneumoniae (mrkF) but also by *E. coli* (FimE). LPS virulence factor (UTP-glucose-1-phosphate uridylyltransferase) was correlated with large adenoma and it was previously confirmed to be significantly increased after hypoxia, which is a hallmark of the tumor microenvironment [26].

Early-stage CRC was associated with different Fimbriae (mrkD, mrkF, and mrkC) and with LPS (dTDP-glucose 46-dehydratase). At this stage, a significant association was revealed between early-stages CRC and RcsB. This transcriptional regular was earlier related to biofilm formation and cell division [27]. Moreover, we noticed a correlation between stages I and II of CRC and EntB (siderophore). Different types of virulence factors have been found in advanced CRC. In fact, several Fimbriae were detected such as FimE, FimB, FimG, erpC, and FimA, which have been shown to enhance the ability of uropathogenic *E. coli* (UPEC) to elicit symptomatic infection [28]. LPS of the outer membrane A and UTP-glucose-1-phosphate uridylyltransferase subunit GalF were also highly detected in these stages. We noticed a significant association between stages III and IV, and hasB, which is involved in capsular polysaccharide biosynthesis and plays an important role in the formation of hyaluronic acid (HA), thus promoting the cellular growth and migration of physiological cells [29]. Additionally, advanced CRC has been correlated with the immunomodulatory protein TcpC, known to counter the host’s innate immune defense by abrogating the function of MyD88 in macrophages that further impairs the involved signaling processes [30]. We also noticed that many types of secretion systems could be involved in the advanced level of adenoma–carcinoma sequence: the type 3 secretion ESPL1 and ESPX1 (type III secretion systems are used by pathogenic species to inject toxins into attacking immune cells [31,32]), and general secretion system D, F, K (general secretion pathway can translocate the small-signal sequence-containing proteins into the periplasm which can be targeted then to ABC transporter/TolC-based export system such as the enterotoxins produced by *E. coli* [33]). We also observed that siderophores were the most detected VFs in early-stage CRC, which is in line with a previous study highlighting the enrichment of entC in CRC [34].

Enterobactin (EntF, FepC, FepG, EntD, EntE, FepB, FepD, Fes, EntS), which was highly detected in advanced-stage CRC stool, has been previously studied and shown to play a role in promoting a pro-inflammatory response, as well as reducing basal reactive oxygen species (ROS) generation in Intestinal epithelial cells [35]. Similarly, yersiniabactin (ybtE and fyuA), which is a major bacterial iron-uptake system, has been shown to promote oxidative stress [36] by triggering three signaling pathways that have been reported to be closely linked to CRC development [37]: the Janus kinases (JAK)/signal transducer and activator of transcription proteins (STAT), Wnt/β-catenin, and PI3K/AKT pathways. However, no previous study has confirmed an association between aerobactin (iucB) and CRC.

While the comparison of the different CRC stages with healthy stools was for prognostic purposes, we compared the different stages with each other to study the variation and the evolution of the virulome throughout the adenoma–carcinoma sequence. No significant differences were revealed between the VFs in small adenoma vs. large adenoma and large adenoma vs. early CRC. However, when we compared adenomas (large and small) with early-stage CRC, we noticed a significant correlation with VFs, such as Fimbriae (SfaG, SfaS), siderophore (EntB), transcriptional regulator (rcsb), immunomodulatory protein (TcpC), a type VI secretion system (which plays an important role in delivering effectors into cells using a bacteriophage-like puncture mechanism) [38], CNF1 (toxins induce the activation of Rho GTPases that triggers different cellular events such as NF-kB activation (via the Akt/IκB kinase pathway) and provide protection against apoptosis) [39], and polysaccharide lyase produced by E. faecalis, involved in malignant transformation, invasion and metastasis [40].

We also compared the virulomes of early- and advanced-stage CRC, and found that stages III and IV were more enriched with Fimbriae (FimB, FimE, fumH, FimI and FimG), transcriptional regulator (Ybta), outer-membrane A (Lps) and siderophores (aerobactin (iucb), enterobactin (fepC and fepG) and yersiniabactin (Ybtp, Irp1, and Irp2).

A significant association between colibactin (clbK, clbF, clbI, and clbL) and advanced-stage CRC was observed. Colibactin is a genotoxic metabolite that causes DNA double-strand breaks in mammalian cells, leading to chromosomal abnormalities, cell cycle arrest, and senescence [41]. It has been demonstrated that colibactin-producing bacteria can promote the emergence of senescent cells by regulating the tumor microenvironment, which can also influence tumor progression and cancer progression through the secretion of growth factors [42]. In addition, E. faecalis produces a fibrinogen-binding protein (FSS2) at this level involved in the bacteria–host interaction and which is necessary in the early stages of infection [43].

The VFs involved in cell adhesion were detected in all stages of adenoma and carcinoma; however, binding activity, signal transduction, transcription regulation, and the activity of porins and transporters involved in antibiotic resistance were not found in the adenoma. Chuanfa et al. showed that the gut microbiota of CRC patients had higher antibiotic-resistance genes than those of healthy subjects [44]. The VFs associated with advanced-stage CRC showed a unique functional profile characterized by oxidoreductase activity and Nad metabolism. In addition, biofilm development, previously shown to improve epithelial permeability, promotes the inflammation of pro-carcinogenic tissues via the direct access of bacterial antigens/mutagens to an unprotected epithelial surface [45]. As for pathway analysis, many VFs correlated with advanced-stage CRC are involved in the biosynthesis of nonribosomal siderophore group peptides that could drive colitic and pro-carcinogenic responses [46].

Here, we investigated the network pathogenicity of VFs associated with advanced-stage CRC via multivariate analysis, and noticed that isochorismate synthase was highly correlated with enterobactin exporter. This link was previously described, and isochorismate was considered as a precursor for the biosynthesis of the siderophore enterobactin [47]. In fact, pathogenic bacteria producing isochorismate can uptake iron from other living organisms even when iron concentrations are extremely low [48]. Since iron is required to maintain immune functions, its deficiency may increase oncogenicity by altering neoplastic changes in immunosurveillance and potentially remodeling the tumor immune microenvironment [49]. This could trigger other pathways involved in cell cycle arrest and apoptosis [50].

## 5. Conclusions

In this study, we analyzed the virulome using metagenomic data available in ENA and identified a significant enrichment of predicted VFs in CRC patients’ microbiomes. These VFs were produced by pathogens such as pathogenic *E. coli*, *K. pneumoniae*, *Enterococcus* sp., *Shigella* sp., etc.

Colibactin and siderophores were remarkably more abundant in advanced stages of CRC compared to healthy microbiomes and early stages of CRC. Using this combined approach, our work offers a holistic exploration of the predicted pathogenic potential of the microbiome and reveals specific tax/functions as potential targets in this disease. Further functional analyses are needed to understand the role of VFs in oncogenicity.

## Figures and Tables

**Figure 1 microorganisms-10-02365-f001:**
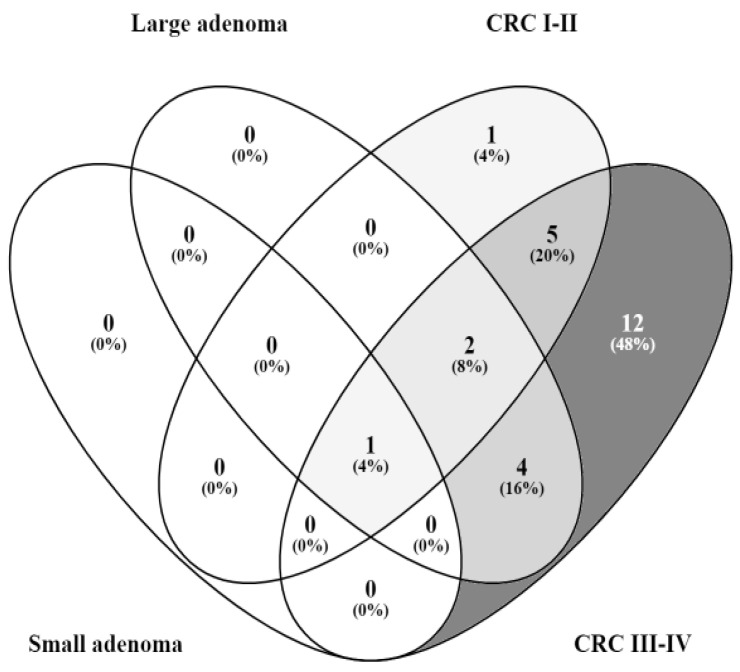
Venn diagram showing the number of shared and unique functions of targeted VFs detected in different stages of CRC.

**Figure 2 microorganisms-10-02365-f002:**
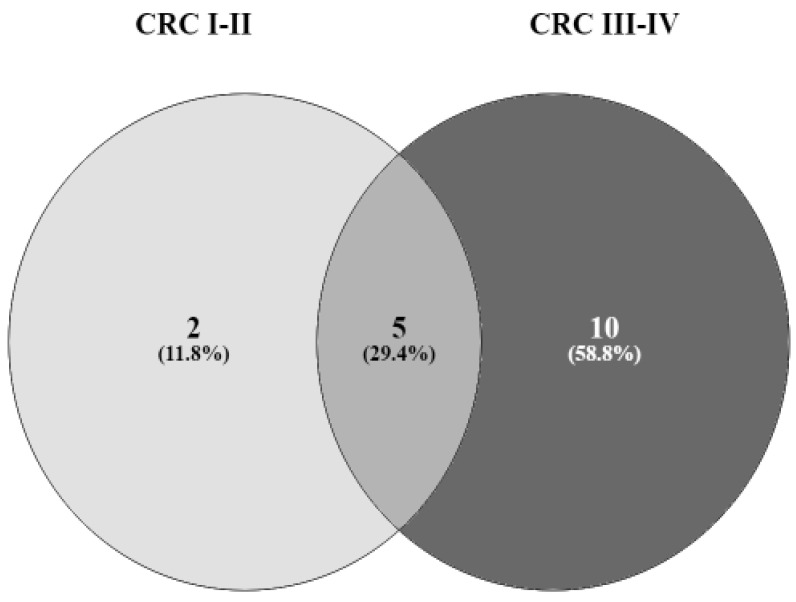
Venn diagram showing the number of shared and unique pathways that include targeted VFs detected at different stages of CRC.

**Table 1 microorganisms-10-02365-t001:** Multivariate analysis of virulence factors detected in advanced stages of CRC.

Coefficients ^a^					
Model	Unstandardized Coefficients		Standardized Coefficients	T	Sig.
	B	Std. Error	Beta		
(Constant)	−0.005	0.008		−0.550	0.584
Type 1 Fimbriae Regulatory protein FimB	0.014	0.057	0.014	0.239	0.812
Type 1 Fimbriae Regulatory protein FimE	0.017	0.040	0.017	0.425	0.672
Type-1 Fimbrial protein, A chain precursor	−0.081	0.032	−0.080	−2.510	0.014
iron-enterobactin ABC transporter permease	0.501	0.053	0.505	9.498	0.000
phosphopantetheinyl transferase component of enterobactin synthase multienzyme complex	−0.072	0.034	−0.075	−2.139	0.036
FimG protein precursor	0.067	0.030	0.069	2.211	0.030
ferrienterobactin ABC transporter periplasmic binding protein	0.041	0.038	0.042	1.092	0.278
*E. coli* common pilus usher EcpC	0.354	0.060	0.351	5.881	0.000
enterobactin/ferric enterobactin esterase	−0.038	0.081	−0.039	−0.468	0.641
enterobactin exporter, iron-regulated	0.213	0.054	0.216	3.910	0.000
general secretion pathway protein D	0.078	0.035	0.074	2.207	0.030
general secretion pathway protein K	−0.111	0.035	−0.109	−3.146	0.002
aerobactin synthesis protein *Iuc*B	0.020	0.022	0.015	0.879	0.382

^a^. Dependent Variable: isochorismate synthase 1.

**Table 2 microorganisms-10-02365-t002:** Functions and pathway analysis of the VFs associated with different stages of CRC.

	Early Stage CRC	Advance Stage CRC
Functions of Virulence factors	Biosynthetic Process Metabolic Process Binding Activity Signal Transduction Regulation of Transcription Cell Adhesion Porin Activity Pilus Assembly Transportation	Metabolic process Ligase activity Binding activity Biosynthetic process Dna integration Dna recombination Cellular response to dna damage stimulus Cell adhesion Nad metabolic process Signal transduction Regulator activity Protein phosphopantetheinylation Biofilm development Host cell recognition Mechanosensory behavior Transportation Hemeostasis process Porin activity Detection of virus Viral entry into the host cell Response to an antibiotic oxidoreductase activity Regulation of transcription Signaling receptor activity
Pathway of virulence factor	Streptomycin biosynthesis O-antigen nucleotide sugar biosynthesis Metabolic pathways Biosynthesis of secondary metabolites Biosynthesis of nucleotide sugars Biosynthesis of siderophore group Nonribosomal peptides Biosynthesis of secondary metabolites two-component system	Biosynthesis of secondary metabolites Biosynthesis of siderophore group nonribosomal peptides Abc transporters Ubiquinone and terpenoid-quinone biosynthesis Metabolic pathway Bacterial secretion system Pentose and glucuronate interconversions ascorbate and aldarate metabolism Amino sugar and nucleotide sugar metabolism o-antigen nucleotide sugar biosynthesis biosynthesis of cofactors Biosynthesis of nucleotide sugars Pentose and glucuronate interconversions galactose metabolism Starch and sucrose metabolism

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
