# Peer review of "Virulence Factors in Colorectal Cancer Metagenomes and Association of Microbial Siderophores with Advanced Stages"

_microorganisms, 2022, doi:10.3390/microorganisms10122365_

Round 1

Reviewer 1 Report

The manuscript entitled “Virulence factors in colorectal cancer metagenomes and association of microbial siderophores with advanced stages” is significant in this field of interest. The manuscript is well-structured with enough data. However, this manuscript has minor issues needed to be addressed. Thus I recommend this manuscript for minor revision.

Kindly check the digits in table 1 and correct it all.

Table -2 alignment confuses the readability its content. Kindly modify it.

Kindly Check entire MS for typographical and format errors.

Eg. process in Large adenoma, CRC I -II, and 171 CRC III-IV

Eg. pathogenic E. coli, K. pneumoniae, Enterococ- 308 cus sp., Shigella sp., etc. Colibactin and siderophores

As MDPI (toxins) journals are open access, most readers use softcopy of articles to read; hence the color figures will increase the and clarity of the figure. Thus if possible, kindly replace all figures (with color one).

Author Response

Comments

Kindly check the digits in table 1 and correct it all.

Table -2 alignment confuses the readability its content. Kindly modify it.

Kindly Check entire MS for typographical and format errors.

Eg. process in Large adenoma, CRC I -II, and 171 CRC III-IV

Eg. pathogenic E. coli, K. pneumoniae, Enterococ- 308 cus sp., Shigella sp., etc. Colibactin and siderophores

As MDPI (toxins) journals are open access, most readers use softcopy of articles to read; hence the color figures will increase the and clarity of the figure. Thus if possible, kindly replace all figures (with color one).

Response

We agree with you, please provide your response in the revised version of the article (in red in pages 3; 4; 5; 7 and 9 ) "Please see the attachment.":

Table 1 has been corrected according to your suggestion.

Table 2 has been modified according to your suggestion.

Entire MS has been checked for typographical and format errors.

All figures has been replaced (with color one).

Reviewer 2 Report

This metagenome study is overall well-described and this enables to enlighten the role of gut-microbes and their virulence factors. I have a few minor comments below;

Detailed comments

Abstract – well prepared and informative, no critical comments.

Introduction – short, but well prepared.

Line 53: Indeed, Pathogenic -> Indeed, pathogenic

Materials and methods

Line 102-103: Please provide information of software, e.g. company, city, country.

P -> p

Results

1)      Overall, change the scientific name of the microbiome to italics. Please check again the name of pathogens throughout the manuscript.

E.coli -> E. coli

2)      p -> p

3)      Supplementary data 3 seemed to be quiet important, thus it may be better to modified into Table. Table can be made with the only significant data of Supplementary data 3.

4)      It may be better to change previous Table 1 according to Journal style.

5)      Table 2 and Table 3 seemed to be less significant. It may be combined into the single Table.

6)      Lines 169-181: These findings should be rephrased within a single paragraph.

7)      Lines 199-209: These findings should be rephrased within a single paragraph.

8)      line 275: Some mistypo; yersiniabactin (Ybtp, Irp1, Irp2)).-> yersiniabactin (Ybtp, Irp1, Irp2).

Discussion – well prepared.

Conclusions – well prepared.

References – should be modified according to Journal style

Author Response

Comments

This metagenome study is overall well-described and this enables to enlighten the role of gut-microbes and their virulence factors. I have a few minor comments below;

Detailed comments

Abstract – well prepared and informative, no critical comments.

Introduction – short, but well prepared.

Line 53: Indeed, Pathogenic -> Indeed, pathogenic

Materials and methods

Line 102-103: Please provide information of software, e.g. company, city, country.

P -> p

Results

1)      Overall, change the scientific name of the microbiome to italics. Please check again the name of pathogens throughout the manuscript.

E.coli -> E. coli

2)      p -> p

3)      Supplementary data 3 seemed to be quiet important, thus it may be better to modified into Table. Table can be made with the only significant data of Supplementary data 3.

4)      It may be better to change previous Table 1 according to Journal style.

5)      Table 2 and Table 3 seemed to be less significant. It may be combined into the single Table.

6)      Lines 169-181: These findings should be rephrased within a single paragraph.

7)      Lines 199-209: These findings should be rephrased within a single paragraph.

8)      line 275: Some mistypo; yersiniabactin (Ybtp, Irp1, Irp2)).-> yersiniabactin (Ybtp, Irp1, Irp2).

References – should be modified according to Journal style

Response

Thank you for reviewing our manuscript. your propositions were all considered and appreciated. Please provide your response in the revised version of the article (in turquoise in pages 2; 3; 5 and 9 ) "Please see the attachment."

For point N° 3: In supplementary data 3 we tried to highlight the most significant result, and we have put only significant correlations with p < 0.05.

For the point N° 6 and N°7: Your proposition was appreciated, however, it made us revise the format in which we wrote the repetitive paragraph, in points 6 and 7 the repeated lines are the legends of figures 1 and 2, they are not in the text/manuscript.

For the point *References: Your proposition was appreciated and errors were corrected appropriately.  

The other points / errors were corrected appropriately. See changes in turquoise.

Reviewer 3 Report

1. Please correct the typo.

line 295; here -> Here

line 122, 131, 261, 272, 306; virolome -> virulome 

2. Please use the correct binomial nomenclature and gene nomenclature.

Genus species, G.species, G.species -> Genus species or G. species

GeneA -> GeneA

3. Please use the correct abbreviation.

line 34; non-polyposis forms (HNPCC) -> hereditary nonpolyposis colorectal cancer (HNPCC)

line57; enterotoxigenic Bacteroides fragilis (BFT) -> enterotoxigenic Bacteroides fragilis (ETBF)

line58; Fusobacterium adhesin A (Fad A) -> Fusobacterium nucleatum adhesin A (FadA)

line63; cytotoxic necrotizing factor 1 (CNF) -> cytotoxic necrotizing factor 1 (CNF1)

Author Response

Comments

  1. Please correct the typo.

line 295; here -> Here

line 122, 131, 261, 272, 306; virolome -> virulome 

  1. Please use the correct binomial nomenclature and gene nomenclature.

Genus species, G.species, G.species -> Genus species or G. species

GeneA -> GeneA

  1. Please use the correct abbreviation.

line 34; non-polyposis forms (HNPCC) -> hereditary nonpolyposis colorectal cancer (HNPCC)

line57; enterotoxigenic Bacteroides fragilis (BFT) -> enterotoxigenic Bacteroides fragilis (ETBF)

line58; Fusobacterium adhesin A (Fad A) -> Fusobacterium nucleatum adhesin A (FadA)

line63; cytotoxic necrotizing factor 1 (CNF) -> cytotoxic necrotizing factor 1 (CNF1) 

Response

Thank you for reviewing our manuscript. your propositions were all considered and appreciated. Please provide your response in the revised version of the article (in green in pages 1; 2; 3; 4; 7; 8 and 9) "Please see the attachment."
